# Porcine *Lawsonia intracellularis* Ileitis in Italy and Its Association with Porcine Circovirus Type 2 (PCV2) Infection

**DOI:** 10.3390/ani13071170

**Published:** 2023-03-26

**Authors:** Giulia D’Annunzio, Fabio Ostanello, Luisa Vera Muscatello, Massimo Orioles, Barbara Bacci, Niccolò Jacumin, Giorgio Leotti, Nicola Tommasini, Giovanni Loris Alborali, Andrea Luppi, Denis Vio, Luciana Mandrioli, Giuseppe Sarli

**Affiliations:** 1Dipartimento di Scienze Mediche Veterinarie, Università di Bologna, 40126 Bologna, Italy; 2Istituto Zooprofilattico Sperimentale della Lombardia e dell’Emilia-Romagna, 25124 Brescia, Italy; 3Dipartimento di Scienze Agroalimentari, Ambientali e Animali, 33100 Udine, Italy; 4Boehringer Ingelheim Animal Health Italia SpA, 20139 Milano, Italy; 5Istituto Zooprofilattico Sperimentale delle Venezie, 33030 Udine, Italy

**Keywords:** *Lawsonia intracellularis*, proliferative enteropathy, diagnosis, PCV2, immunohistochemistry, IHC, swine, Italy

## Abstract

**Simple Summary:**

In swine, the diagnosis of enteric diseases is challenging due to simultaneous presence of one or more microbic agents sharing similar clinical signs and pathological lesions. Therefore, the study of microscopic lesions is crucial in establishing the role of pathogens as causative agents in cases of co-infections; this can be confirmed by demonstrating agent–lesion co-localization. Although PCR is more sensitive than IHC, the latter provides a positive result only when the amount of antigen is significant, and therefore is likely the cause of the enteric pathology. *Lawsonia intracellularis* is the aetiologic agent responsible for Porcine proliferative enteropathy. However, it is complex to determine whether this intracellular bacterium is the cause of clinical disease due to its high prevalence in the field. In this study, we investigate the agreement between PCR and IHC results for *L. intracellularis* evaluation, and the infection and co-infection by porcine circovirus type 2 and *L. intracellularis* in the ilea of pigs presented with suspected proliferative enteropathy in Italy.

**Abstract:**

The objective of this study was to employ a diagnostic algorithm, which involves detecting positive farms by stool PCR followed by PCR and histology/immunohistochemistry on ileum samples, for diagnosing *Lawsonia intracellularis* proliferative enteritis in Northern Italy. The primary aim was to examine the relationship between the gold standard of *L. intracellularis* diagnostics, namely histology and immunohistochemistry, and PCR in acute and chronic cases of *L. intracellularis* enteritides. An additional goal was to investigate the coinfection of *L. intracellularis* with porcine circovirus type 2 (PCV2). Twenty-eight ileum samples, including four from acute cases and 24 from chronic cases, were collected. PCR yielded positive results in 19 cases (four acute and 15 chronic cases). In comparison, immunohistochemistry was positive in 16 cases (four acute and 12 chronic cases), with an observed agreement of 89%. The findings suggest that performing the two tests in series can increase the specificity of the causal diagnosis. PCR may be used as a screening tool to identify the presence of the microorganism, and only positive cases will be examined by histology and immunohistochemistry to confirm the causative role of *L. intracellularis*. Co-infection with PCV2 was demonstrate in two out of four acute cases and in two out of 24 chronic cases, providing further evidence to support the hypothesis that when the infection starts with ubiquitous pathogens such as *L. intracellularis*, it may boost the possibility of PCV2 replication, especially in acute cases. As a result, this may trigger a transition from subclinical to clinical forms of PCV2 disease.

## 1. Introduction

Enteric diseases pose a significant challenge in intensive pig farming, not only causing production and economic losses but also impacting animal welfare and necessitating antimicrobial use. These on-farm enteric diseases often involve multiple infections, leading to complex clinical patterns that make it difficult to identify the role of each agent and implement effective control measures [1]. 

Post-weaning, enteric diseases in pigs may be caused by bacteria such as enterotoxigenic *Escherichia coli* (ETEC) or Shiga toxin-producing *E. coli* (STEC). Additionally, viral and parasitic pathogens such as rotavirus, cryptosporidia, and coccidia play a role in the first weeks of life. In the growing and finishing phases, the most significant diseases include salmonellosis, porcine dysentery by *Brachyspira hyodysenteriae*, spirochetosis by *Brachyspira* spp., and proliferative enteropathy (PE) by *Lawsonia intracellularis* and ascaridiosis [1]. 

*L. intracellularis* is a Gram-negative bacterium that has been identified as the primary causative agent of enteritides localized in the aboral small intestine of swine, specifically in the ileum [2]. This condition is referred as “proliferative enteropathy” (PE). In the acute haemorrhagic form of the disease (proliferative haemorrhagic enteropathy, PHE), the PE is characterized by thickening and lifting into folds of the mucosa of the ileum, which assumes the so-called “cerebroid” appearance. This is associated with mucosal haemorrhage and blood clots in the lumen. In the chronic form, known as the porcine intestinal adenomatosis (PIA), the thickening of the mucosa is the main feature. Necrotic enteritis (NE) is a less frequent form of the disease and is characterized by mucosal necrosis and fibrin exudation due to complication by other bacteria. Its chronic evolution leads to mural fibrosis, known as regional ileitis or “hosepipe guts” [3,4,5]. In addition to the acute (PHE) and chronic (PIA) clinical presentations of PE, subclinical forms of infection must also be considered. These are the most common and subtle forms to diagnose, since diarrhoea is absent, and the only obvious signs are a reduction in average daily weight gain (ADG), growth dissimilarity in groups and an increase in cast-off animals [3,4,6]. It is worth noting that the macroscopic features of PE, primarily haemorrhage and necrosis, are shared with other swine enteritides. Therefore, the final diagnosis of PE caused by *L. intracellularis* requires histologic confirmation of the typical lesion, which is represented by the hyperplasia of intestinal crypts. In addition, its co-localization with the presence of the antigen through the immunohistochemistry or immunofluorescence or in situ hybridization [5,7] is also required for a definitive diagnosis.

Porcine circovirus-associated diseases (PCVDs) can manifest as both the systemic form of porcine circovirus type 2 (PCV2) disease and the clinical intestinal form, which is referred to as PCV2-enteric disease, PCV2-ED. Both forms of enteritis can be clinically apparent with diarrhoea [1,8]. *L. intracellularis* and PCV2 infections are equally associated with growth retardation, emaciation, increased mortality, and diarrhoea in weaned pigs. In addition, these diseases are characterized by lesions that are sometimes macroscopically indistinguishable, such as necrotizing ileitis and colitis [2]. Therefore, especially when considering co-infections, the study of microscopic lesions is a critical tool to confirm the role of a pathogen as the causative agent of the lesion by demonstrating agent–lesion co-localization [2]. 

The association between PCV2 and *L. intracellularis* in the development of enteric diseases in swine has been established through field investigations conducted in Denmark [2] and in Estonia [9]. Furthermore, PCV2 and *L. intracellularis* have been used experimentally to induce PCV-2 associated enteritis in swine [10]. Both pathogens have been contextually identified in wild boars in Brazil [11].

The purpose of this study is to compare histology and immunohistochemistry (IHC) investigations for the diagnosis of *L. intracellularis* enteritis with PCR results on ileum of pigs submitted to the laboratories of the Istituto Zooprofilattico Sperimentale delle Venezie (IZSVe), and the Istituto Zooprofilattico Sperimentale della Lombardia e dell’Emilia-Romagna (IZSLER), for suspected *L. intracellularis* proliferative enteropathy, in Italy. Furthermore, this study aims to demonstrate the co-infection *L. intracellularis*-PCV2 in situ through immunohistochemistry. 

## 2. Materials and Methods

### 2.1. Sampling

The tissue specimens examined are part of a case series collected to define the relationship between the diagnostic gold standard for *L. intracellularis* (i.e., histology and immunohistochemistry) and PCR in both acute and chronic cases of *L. intracellularis* enteritides. A total of 24 farms were enrolled based on anamnesis, which revealed the presence of *L. intracellularis,* as confirmed by a positive PCR test conducted on stool samples. The selection process for these farms was based on specific criteria, which included the gross appearance of either acute/severe (haemorrhagic/necrotic) or chronic (proliferative) forms of enteritides. Figure 1 serves as a visual aid for identifying these gross lesions.

All samples in the study were taken from animals found dead on the farm. No animals were sacrificed for the purpose.

In order to collect samples, the terminal portion of the ileum located near the ileocecal valve was chosen as it was deemed the most appropriate site for *L. intracellularis* detection. The selected tract of ileum was formalin-fixed to preserve the samples stored at ambient temperature to maintain their integrity.

Histopathology and immunohistochemistry have been investigated at the Pathology Service of the Department of Veterinary Medical Sciences (DIMEVET), University of Bologna, Italy, while the biomolecular tests (PCR or RT-PCR) for *L. intracellularis*, *Brachyspira hyodysenteriae*, and *B. pilosicoli* on stool pools and, only for *L. intracellularis*, also on gut wall were performed at the laboratories of the Istituto Zooprofilattico Sperimentale delle Venezie (IZSVe, Udine, Italy) and the Istituto Zooprofilattico Sperimentale delle Lombardia e dell’Emilia-Romagna (IZSLER, Brescia and Parma, Italy).

### 2.2. Isolation of Brachyspira hyodysenteriae and Brachyspira pilosicoli 

Faecal samples were incubated on blood agar culture plates supplemented with rifampin, vancomycin and colistin for 5 days at 37 °C in an anaerobic atmosphere. Evaluation of their hemolytic characteristic (strong and weak hemolysis) allowed the samples to be classified as suspected *B. hyodysenteriae* or *B. pilosicoli*, according to Gasparrini et al. [12]. Bacterial cells were then collected and resuspended in 200 µL of DNase/RNase-free water, and DNA was obtained by boiling at 98 °C. 

A species-specific real-time PCR confirmed the identification of *B. hyodysenteriae* or *B. pilosicoli*. The primers used are reported in Table 1. 

### 2.3. Real-Time PCR to Brachyspira hyodysenteriae, Brachyspira pilosicoli and Lawsonia intracellularis

Preliminarily, stool or intestinal scrapings samples were diluted 1:10 in phosphate-buffered saline, PBS (1 g of stool in 9 mL PBS). DNA was extracted from 200 µL of a diluted sample using a MagMAX CORE Nucleic Acid Purification Kit (Applied Biosystem, Monza, Italy). The total volume for each assay was 25 µL and consisted of: 12.5 µL 2x QuantiNova Probe PCR masterMix (Qiagen, Milano, Italy), 1.25 µL of each primer (final concentration 0.5 µM), 0.5µL of probe (final concentration 0.2 µM), 2.2 µL of water, 2 µL of EXO IPC mixture, 0.3 µL of Exo IPC DNA (TaqMan Exogenous Internal Positive Control Reagents, Applied Biosystem) and 5 µL of extracted DNA. 

Real-time PCR was performed on a CFX-96 System (Bio-Rad, Segrate, Italy). According to Willems and Reiner [13], the probes (Table 1) were all FAM-labeled, and the reaction program was as follows: activation of DNA polymerase at 95 °C for 2 min, 40 cycles with denaturation at 95 °C for 3 s and annealing and extension at 60 °C for 30 s.

### 2.4. Histology and Immunohistochemistry (IHC)

Tissue samples were fixed in 10% buffered formalin and embedded in paraffin. From each ileum specimen, 3-micron-thick sections were obtained, one stained with hematoxylin-eosin, while the other two underwent immunohistochemical staining for *L. intracellularis* and PCV2. Details of the immunohistochemical protocol used are provided in Table 2. The antigen–antibody reaction was revealed using diaminobenzidine (0.04% for 2 min).

In each immunohistochemical run, a section of ileum previously identified as positive for the presence of *L. intracellularis*, and a section of lymph node from a PCV2-SD case positive to PCV2 by both PCR and IHC were used as positive controls.

The specificity of the immunohistochemical stain has been validated by adding, as the primary reagent, an antibody of the same isotype as the primary antibodies (IgG1) but of irrelevant specificity.

A proposed grading system was used to assess the histologic severity of enteritis and crypts hyperplasia [10]. First, each section of ileum was scored for the presence of inflammatory infiltrate in the mucosa (0, normal; 1, mild cellular infiltrate predominantly lymphohistiocyte-like; 2, moderate infiltrate with submucosa involvement; 3, severe cellular infiltrate of mucosa and submucosa) and hyperplasia of intestinal crypts (0, normal; 1, mild crypt hyperplasia; 2, moderate hyperplasia; 3, severe crypt hyperplasia with or without crypt herniation in the submucosa). In addition, lymphocyte depletion was graded in the Peyer’s patches according to Opriessnig et al. [15] (0, normal; 1, depletion of Peyer’s patches; 2, depletion of Peyer’s patches and replacement with histiocytic infiltrate; 3, depletion, replacement with histiocytic infiltrate and presence of multinucleated cells).

The Immunohistochemical reaction for *L. intracellularis* was considered positive in case of brown-coloured granular material in the apical cytoplasm of intestinal crypt cells, and in the cytoplasm of macrophages in the interstitium of intestinal mucosa (Figure 2). The immunohistochemical stain was assessed by a semiquantitative score from 0 to 3 [2,16] (0, no signal; 1, focal signal; 2, moderate multifocal signal; 3, extensive signal) based on the amounts and extent of the specific stain. 

For PCV2, the test was scored positive in case of brown staining of the cytoplasm of follicular dendritic cells in Peyer’s patches and/or macrophages in the lamina propria of the mucosa, applying a score from 0 to 3 [10,15] (0, negative; 1, cells with PCV2 antigen staining in less than 10% of lymphoid follicles; 2, cells with PCV2 antigen staining in 10–50% of lymphoid follicles; 3, cells with positive staining for PCV2 antigen in more than 50% of lymphoid follicles).

### 2.5. Other Investigations

The results of bacteriological investigations (*Brachyspira hyodysenteriae*, *Brachyspira pilosicoli* and *Lawsonia intracellularis*) for each out of the 28 animals are reported in Appendix A.

Agreement between PCR and IHC results for *L. intracellularis* was assessed using the Cohen’s Kappa (*K*) coefficient [17].

## 3. Results

The present caseload encompasses 28 ileum samples collected from 24 farms. One farm provided four samples, another provided two samples, and the remaining 22 each provided one sample. Two cases (ID 1 and 2 in Appendix A) exhibited acute clinical disease and a macroscopic feature of acute haemorrhagic enteritis (PHE), while the other two cases (ID 3 and 4 in Appendix A) demonstrated clinical indication of acute disease but histologically displayed necrotic enteritis (NE). The remaining 24 cases were characterized as suspected chronic clinical forms (PIA) and involved pigs with a history of poor growth and diarrhoea.

Two out of the four pigs with the clinical acute form were under 100 days old. The remaining 24 animals showed the chronic form; of these, 50% were younger than 100 days old (Appendix A).

The results of histological grading of enteritis and lymphocyte depletion are reported in Appendix A. The main findings indicate severe enteritis with a high degree of crypt hyperplasia, which was more pronounced in acute cases (median value of inflammatory infiltrate was 3 on a 0–3 scale) than in chronic cases (median value of 2.5). Peyer’s patches showed severe lymphocyte depletion in acute cases (median value of 2 on a 0–3 scale), but only mild or undetectable depletion was observed in chronic cases (median value of 0.5).

Immunohistochemistry confirmed the presence of *L. intracellularis* in all four cases of acute haemorrhagic/necrotic enteritis (Figure 2 and Figure 3). On microscopic examination, the mucosa of the ileum appeared severely infiltrated by numerous lymphocytes, plasma cells and histiocytes intermixed with extravasated erythrocytes; the mucosa was thickened by moderate/severe hyperplasia of the intestinal crypts, with occasional multifocal crypts herniation in the submucosa (Figure 4A–C). The histopathological examination revealing extensive necrosis affecting the superficial and middle layers of the mucosal lining serves as definitive evidence for the diagnosis of necrotic enteritis, as illustrated in Figure 4D. In one of the two cases of haemorrhagic proliferative enteropathy, lymphocytic depletion and replacement with histiocytic cells in the Peyer’s patches was detected. In addition, the aforementioned case demonstrated a positive result for PCV2 immunohistochemical staining, which was observed in over 10% of lymphoid follicles, as well as in macrophages within the lamina propria. (Figure 5A). In one of the two cases of necrotic enteritis, the presence of PCV2-positive histocytes was also observed in the deep submucosa. Additionally, the affected tissue exhibited a granular brown material within the necrotic mucosa. 

In cases of chronic proliferative enteropathy (Figure 6A), crypt hyperplasia and the degree of inflammatory cell infiltration were predominantly severe/moderate with a median value of 2.5 for inflammatory infiltrate and 2.0 for crypts hyperplasia. However, immunohistochemical staining (Figure 6B) was positive only in 12 out of 24, with a median positivity score of 0.5. In these group of samples, lymphoid depletion was mild to moderate, and only in one case, severe depletion was associated with the presence of multinucleated cells in follicles. PCV2-positive histiocytes in ileum mucosa (Figure 5C,D) and in the centre of follicles (Figure 5B) in Peyer’s patches were detected in two cases (ID 6 and 7 in Appendix A).

Following the diagnostic gold standard criteria for *L. intracellularis* enteritis in swine, the causative role of the bacterium was confirmed in all cases of acute haemorrhagic/necrotic ileitis and in 12 out of 24 chronic ileitis.

PCR for *L. intracellularis* was positive in all four cases of acute disease and in 15 out of the 24 cases of chronic disease. None of the IHC-positive samples were PCR-negative, and all PCR-negative samples were also IHC-negative. The observed agreement between PCR and IHC was 0.89, with a 95% confidence interval (95%CI) ranging from 0.72–0.98. (Table 3). Moreover, the Cohens’ Kappa value of 0.77 indicates a substantial level of agreement between the PCR and IHC results (Table 3). Specifically, when considering histology as the gold standard, 16 out of the 19 PCR-positive samples were confirmed by IHC; three (PCR+/IHC-) of the remaining 12 IHC negative samples belonged to the group of suspected chronic form of disease.

Co-infection of *L. intracellularis* with PCV2 and co-localization of lesion with the relative aetiology (*L. intracellularis* in the cytoplasm of intestinal crypts and PCV2 stain in the centre of lymphoid follicles and multifocally in macrophages in the lamina propria) was objectively assessed in two out of four acute cases and two out of 24 chronic cases.

## 4. Discussion

In the present study, all suspected proliferative enteropathy (PE) cases exhibited hyperplasia of the intestinal crypts. However, the presence of *Lawsonia intracellularis* was detected by immunohistochemistry (IHC) in only 16 out of 28 (57%) examined ilea. Notably, all the samples from the four clinical cases were positive for both IHC and PCR, while only 12 out of the 24 samples from herds with cases of diarrhea and/or suspected subclinical cases were positive to *L. intracellularis* by IHC. Furthermore, three cases exhibited discordant results with the latter group of samples (PCR+/IHC-). Despite the substantial agreement (*K* = 0.77) between PCR and IHC results, PCR demonstrated greater sensitivity in diagnosing *L. intracellularis* infection. This discrepancy could be attributed to the variable amounts of intracellular bacteria present in swine infections, including cases below the limit of detection (LOD) of the immunohistochemical method. Additionally, the sampling strategy that involved analysing one swine in most farms may have limited the probability of detecting cases with detectable antigen levels by IHC. Experimental infection studies have shown that at least five days of intracellular replication are necessary to reach detectable levels of *L. intracellularis* by IHC or in situ hybridization (ISH). The antigen could be detected up to 14 days post-infection [5]. It is plausible that the examination of more than one pig would have increased the probability of detecting cases with antigen detectable by immunohistochemistry. As for the nine cases that tested negative for both *L. intracellularis* PCR and IHC, it is plausible that they represent actual negative cases. Alternatively, due to the segmental distribution of *Lawsonia* lesions [18], the result could be due to analysing only one sample per pig.

Upon comparing the available literature, it has been observed that the percentage of *L. intracellularis*-positive samples by IHC in this particular caseload (57%) is substantially higher than the rates reported in similar field investigations in Denmark (18%) [2], or in Estonia (22%) [9]. This discrepancy could be attributed to the criteria used in this investigation to select farms exhibiting anamnesis indicative of *L. intracellularis,* which may have influenced the outcomes.

Several types of investigation for diagnosing PE in swine are available, each with its strengths and limitations [5]. The clinical presentation alone is inadequate for diagnosis as it overlaps with other intestinal pathogens such as *Brachyspira hyodysenterie, Brachyspira pilosicoli*, *Salmonella enterica*, *Escherichia coli*, and PCV2 [2,5]. The gross features of PE are also shared with PCV2 enteritis [2,10]. However, the lesion may be absent, overlooked when small and focal, or hampered when masked or complicated by other diseases such as salmonellosis or swine dysentery [5]. This explains the high percentage of IHC-negative results if samples are collected only on a macroscopic basis. Moreover, the pathognomonic crypts hyperplasia of *L. intracellularis* infection is not specific, as similar changes can occur in response to mucosa injury or ulceration [5]. Therefore, a combination of histology and unequivocal localization of the causative agent by IHC, immunofluorescence (IF) or ISH is required for accurate diagnosis [5]. PCR has greater sensitivity than IHC and can detect *L. intracellularis ante-mortem* more rapidly and accurately during the first stage of infection [5], providing positive results in a greater proportion of cases than IHC. However, PCR cannot be used to confirm the diagnosis of PE from stool samples alone [18]. Furthermore, it can identify the presence of the *L. intracellularis* genome both in stool and in the intestinal mucosa, even when the amounts of microorganisms are so low that it is unlikely to be the actual cause. In contrast, IHC provides a positive result only when the amount of *L. intracellularis* is significant, and therefore is likely the cause of the enteric pathology. More concordant results can be reached between qPCR and IHC, as a cycle threshold (Ct) less than 20 predicts the presence of lesions and *L. intracellularis* detected by IHC [18].

Upon analysis of the findings of our investigation, it appears that using the IHC examination yields more reliable results in terms of identifying the causal diagnosis of the observed enteric pathology. Therefore, it may be beneficial to perform PCR and IHC tests in series to increase the specificity of the causal diagnosis attributed to *L. intracellularis*; PCR can be used as a screening technique to identify the presence of the microorganism and, in cases of positivity, samples should then undergo further examination by histology and immunohistochemistry for definitive confirmation of *L. intracellularis*’ role as the actual causative agent.

In this investigation, co-infection of PCV2 and *L. intracellularis* was identified in two out of four cases of acute disease and two cases of chronic enteritis, resulting in an overall percentage of four (double-positive) out of 16 IHC *Lawsonia*-positive cases (25%). The emergence of PCV2 and its associated diseases in the first decade of the 21st century has dramatically changed disease patterns in pig farming, highlighting the pathogenic potential of endemic agents as a direct consequence of the immunosuppressive abilities of PCV2 [19]. PCV2 infection is known to impair both innate and adaptive immune responses [20], leading to alterations in intestinal mucosal immunity, such as changes in the expression of immune-related genes, local cellular immune set-up, and IgA production [21]. When a disease is caused by more than just a single pathogen, it is necessary to determine the primary and secondary roles of each. In the case of co-infection with PCV2 and *L. intracellularis*, the literature clearly indicates their respective roles. Recent Danish data show altered infection dynamic and antibody development to PCV2 and *L. intracellularis* in pigs previously exposed to PCV2 [22]. These changes can result in potential dysmicrobism and an increased likelihood of co-infections. The most commonly reported co-infections are between PCV2 and *Salmonella*, *Brachyspira* spp., and/or *L. intracellularis* [2,9,10,23]. Additionally, PCV2 should be considered a primary intestinal pathogen in swine, associated with macroscopic features of necrotic enteritis [2,10]. In their study, Jensen et al. [2] outlined the differences and similarities between *L. intracellularis* and PCV2 enteritis in pigs. *L. intracellularis* enteritis is characterized by crypt hyperplasia, while PCV2 enteritis is marked by histiocytes positive to PCV2 infiltrating the mucosa, sub-mucosa and Peyer’s patches. However, both diseases share features such as depletion and necrosis of Peyer’s patches and monocyte/macrophage infiltration of the lamina propria. Jensen et al. [2] showed evidence of *L. intracellularis*-PCV2 co-infection through the use of immunolabelling in 34 out of 64 intestinal samples with suspected PE. At the same time, only PCV2 was detected in 27 out of the remaining 30 cases of enteritis. The authors used a double immunohistochemical staining to show the co-presence of *L. intracellularis* and PCV2 in macrophages [2]. However, in a separate study by Opriessnig et al. [10], the two pathogens did not share a common cell type or location in pigs experimentally infected with both agents to reproduce PCV2-associated enteritis. As shown by experimental investigations, PCV2 replication is enhanced by co-infection with viruses. Although the exact mechanism of PCV2 potentiation is still unknown, it is supposedly enhanced by other pathogens’ indirect initiation of host cell replication [10]. As demonstrated in field cases by Jensen et al. [2], the co-presence of PCV2-*L. intracellularis* in macrophages can explain why the probability of PCV2 replication and associated disease increases in the intestines of *L. intracellularis*-infected swine. Co-infection with other pathogens or immunomodulation is known to increase the severity of PCV2 infection [2], and the synergy between *L. intracellularis* and PCV2 is not surprising. This can support a secondary role of PCV2 in double infection PCV2-*L. intracellularis*. In the caseload of this study, a higher frequency of acute cases sharing both infections was observed, supporting the hypothesis that when infection by ubiquitous pathogens such as *L. intracellularis* occurs, it contributes to an increased likelihood of PCV2 replication, creating an opportunity to switch from subclinical to clinical PCV2 disease. However, doubts have been raised about the actual existence of PCV2 intestinal and lung diseases [24], as almost all lung and intestinal involvement is associated with systemic lymphoid lesions that lead to a diagnosis of systemic PCV2 disease when other lymphoid organs are available for histology. It is unknown if the PCV2-positive enteritis described in this study represents actual PCVD-enteric disease, as data on other target tissues were unavailable.

## 5. Conclusions

Swine enteric diseases, similar to respiratory pathologies, are complex conditions often characterized by the simultaneous presence of one or more microbial agents. Therefore, while reducing antimicrobial use on farms through prophylactic measures is essential, it is equally important to establish tailored diagnostic protocols that can provide a definitive diagnosis and enable targeted control and prophylaxis. This study’s results on the diagnostic pathway for *L. intracellularis*-induced proliferative enteritis underscore the importance of histopathology as the gold standard diagnostic technique for identifying this condition. The more severe and acute cases of PE may be linked to the activity of pathogens that can influence each other. In the case of *L. intracellularis* and PCV2, both hypotheses of PCV2 as the primary agent causing severe immune system dysfunction, which worsens the pathogenicity of the other agents and the ability of *L. intracellularis* to prime PCV2 replication, should be considered.

## Figures and Tables

**Figure 1 animals-13-01170-f001:**
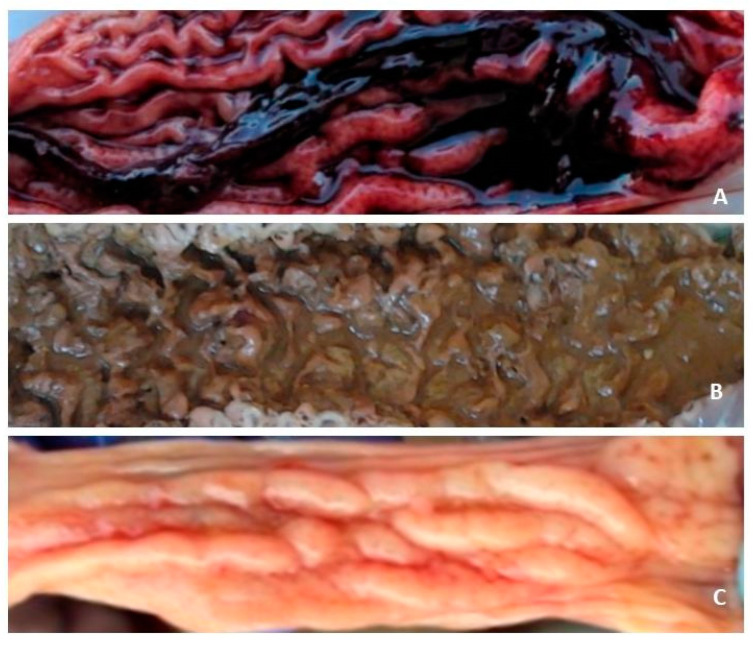
(**A**) Ileum, gross lesion from a clinical case of proliferative haemorrhagic enteropathy (PHE) showing mucosal thickening and reddening associated with luminal blood clots. Out of four cases conferred with anamnesis of acute PE, two, available also as macro photos, confirmed this pattern histologically, while the other two (available only as fixed samples) showed necrotic enteritis (NE), as shown in (**B**). (**C**) From the ileum, a gross lesion of a sample from a pig with chronic diarrhoea, showing mucosal thickening.

**Figure 2 animals-13-01170-f002:**
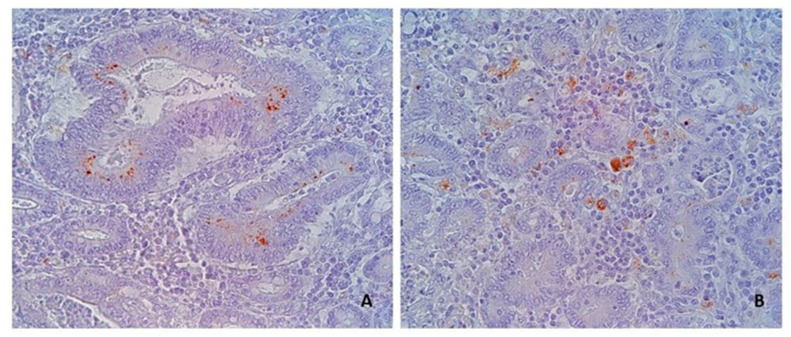
Swine ileum, Case ID 1: Immunohistochemistry to *L. intracellularis* showing a positive stain in the apical cytoplasm of the cells in the crypts (**A**) and in macrophages (**B**) in the mucosal interstitium. (**A**,**B**): 40×.

**Figure 3 animals-13-01170-f003:**
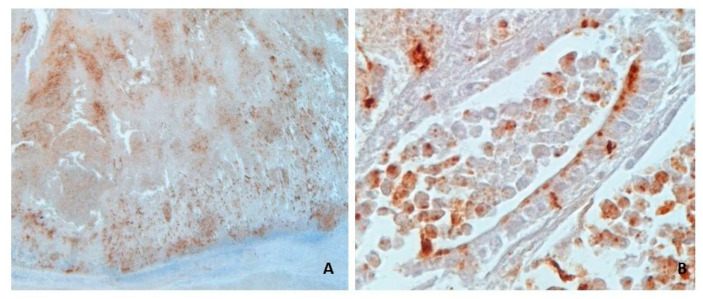
Swine ileum, Case ID 3: (**A**,**B**) Immunohistochemistry to *L. intracellularis* in a case of necrotic enteritis. In (**B**), the immunohistochemical stain is appreciable both in the apical side of crypts cells and in cell debris within the lumen of the crypts. (**A**): 2.5×. (**B**): 63×.

**Figure 4 animals-13-01170-f004:**
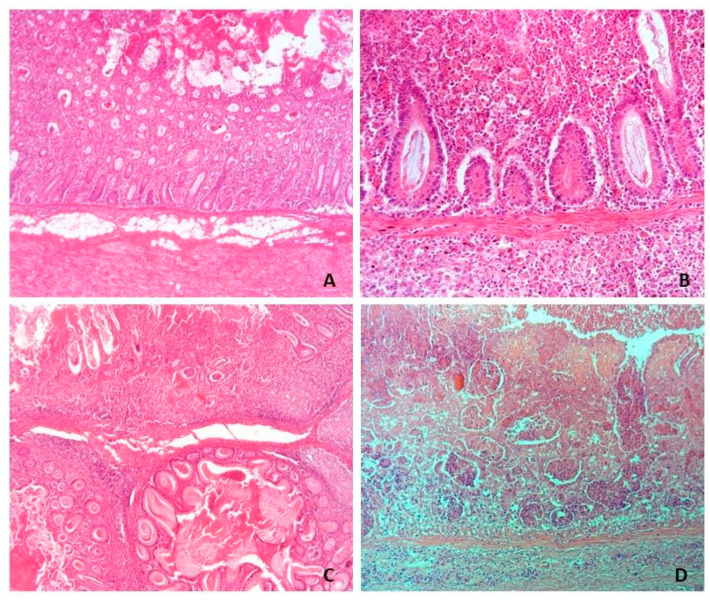
Swine ileum, (**A**,**B**) Case ID 2: Proliferative haemorrhagic enteropathy (PHE). Necro-haemorrhagic content in the lumen is associated with moderate crypt hyperplasia and mucosal expansion by numerous lymphocytes, plasma cells and histiocytes intermixed with extravasated erythrocytes. In (**B**), crypts with absent differentiated (goblet) cells are shown. (**C**) Case ID 1: PHE; Crypt hyperplasia and herniation in the submucosa. (**D**) Case ID 3: Necrotic enteritis (NE); Necrosis of the mucosa above muscularis mucosae and cell infiltration in the submucosa. (**A**,**C**): 6.3×. (**B**): 25×. (**D**): 16×.

**Figure 5 animals-13-01170-f005:**
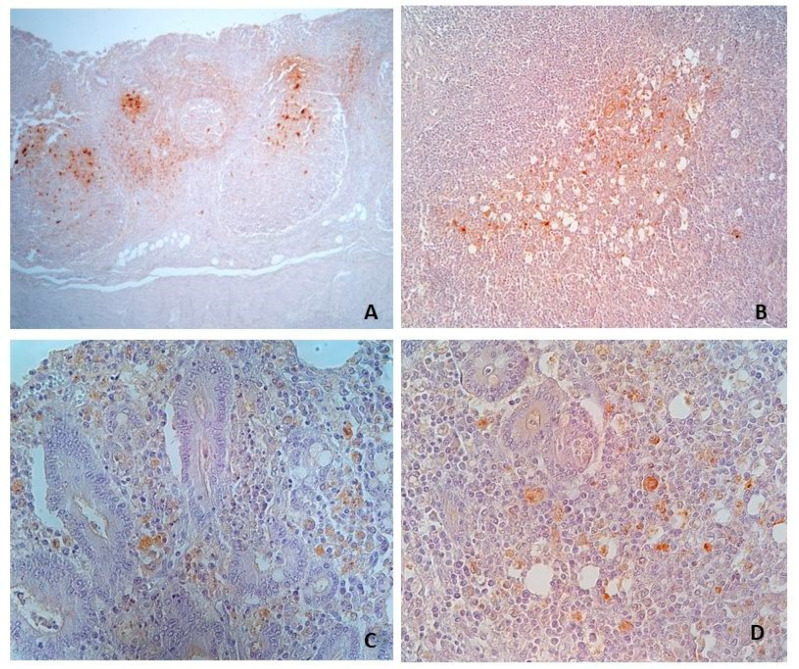
Swine ileum, (**A**) Case ID 1: Depletion in Peyer’s patches and immunohistochemical positive reaction to PCV2 in follicular histiocytes. (**B**,**C**) Case ID 6: (**B**) Positive immunohistochemical stain in the centre of a follicle of the Peyer’s patches, and (**C**) in histiocytes infiltrating the mucosa. (**D**) Case ID 7: PCV2 positive histocytes infiltrating the mucosa. (**A**): 6.3. (**B**): 16×. (**C**,**D**): 40×.

**Figure 6 animals-13-01170-f006:**
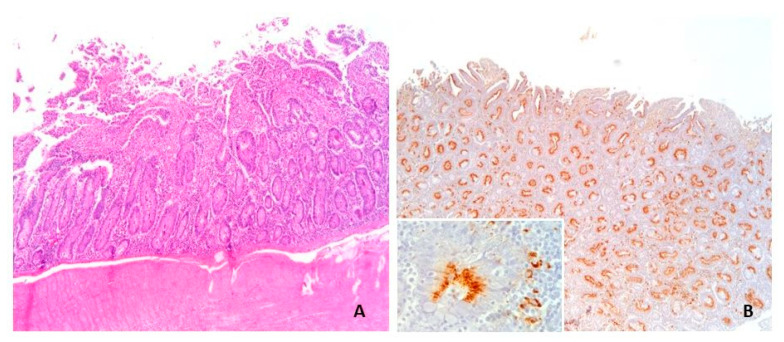
Swine ileum; porcine intestinal adenomatosis (PIA). Case ID 22: (**A**) Moderate hyperplasia of the crypts and (**B**) immunohistochemical positive stain to *L. intracellularis* appreciable (inset) both in the cell crypts and in interstitial macrophages. (**A**,**B**): 6.3×.

**Table 1 animals-13-01170-t001:** Primers and probes used for PCR.

Microorganism	Primers
*Brachyspira hyodysenteriae*	RT-dir: 5′-GAC ATG ATG TTA CTA AAA TAG ACT GGG CT-3′ RT-rev: 5′-CAG GCC AAG AAC CAG TAG CAA G-3′ Probe-5’-(FAM)TTG AAG ACA CTT ACG ATA AAC (MGB)
*Brachyspira pilosicoli*	RT-dir: 5′-GAA GCT ATG CCT AGA GTT ATG GCT AAC-3′ RT-rev: 5′-CCT AAA TGC AAT TCT ATA CCA GCA TC-3′ Probe-5’-(FAM)TTT TGA CAA AGA GAT TAC TGA TGA G (MGB)
*Lawsonia intracellularis*	RT-dir: 5′-TCT CTG CTG CAT GTA ATG AAA TCA-3′ RT-rev: 5′-CTC CTT GAA TAC AAT CCA CAA CAA A-3′ Probe-5’-(FAM) AAA TGG AGA ACT CCT TGA TC (MGB)

**Table 2 animals-13-01170-t002:** Primary antibodies and operating conditions used for immunohistochemistry tests.

Primary Antibody	Clone	Source	Dilution/ Incubation	Antigen Retrieval
*Lawsonia intracellularis*	L. intracellularis Supernatant of A_1_8B_1_ [14]	DTU-VET	1:200/ overnight 4 °C	n.a.*
PCV2	mAb 36A9, Isotype: IgG2a Anti-VP2 PCV2	Ingenasa, Madrid, Spain	1:1500/ overnight 4 °C	30 min at 37 °C in protease XIV 0.05%, pH 7.5

* not applicable.

**Table 3 animals-13-01170-t003:** Observed agreement of IHC and PCR results.

		IHC			Observed Agreement *	Cohen’s Kappa
		Positive	Negative	Total		
**PCR**	**Positive**	16 ^a^	3 ^b^	19	0.89 (95%CI: 0.72–0.98)	0.77 (95%CI: 0.54–1)
	**Negative**	0 ^c^	9 ^d^	9		
	Total	16	12	28 ^n^		

* Observed agreement was calculated as follows: (a + d)/n.

## Data Availability

Not applicable.

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
