# Peer review of "Porcine Lawsonia intracellularis Ileitis in Italy and Its Association with Porcine Circovirus Type 2 (PCV2) Infection"

_animals, 2023, doi:10.3390/ani13071170_

Round 1

Reviewer 1 Report

Please find review comments attached.

Author Response

General Question: As the enteritis disease progression can present as chronic, the acute phase of the infection can be assumed to not be fatal to the pigs? If there is an increased risk of mortality, is this due to L. intracellularis infection directly, or as a combination of a few possible causative agents?

Answer: we thank the reviever for this question. Acute and chronic forms of PE have different pathogenetic mechanisms (absence of inflammatory cells and death of immune cells in chronic disease while increase in the number of macrophages in the mucosa and high level of proinflammatory cytokines in the acute disease (Campillo et al., J. Vet. Diagn. Invest. 2021; https://doi.org/10.1177/10406387211003551)). The reason of this is unknown and we do not know to date if the chronic disease follows an acute form. The death due to PE is reported in acute haemorrhagic disease and in necrotic enteritis this latter considered a complication by other bacteria, mainly Salmonella.  

Lines: 79-82: Can serology be run on these samples e.g. ELISA for L. Intracellularis* Is IHC/histological confirmation the only antigen detecting method? It would be interesting to see if other antigen/antibody-based diagnostics provide a comparable result.

Answer: Serology, by detecting antibodies in blood, gives for sure indication of a previous infection but does not provide any evidence of a still active presence of Lawsonia intracellularis in the gut. In place of immunohistochemistry also in situ hybridization can be used so the diagnostic tool includes always the evidence of the pathogen in tissues highlighting its antigen or nucleic acid. In case you have to decide for IHC or ISH, it is known this latter is more vulnerable to the effects of autolysis compared to IHC. So we decided for a more reliable test.

Line 102-104: Typographical error, no need for commas between "furthermore..." and "...the co- infection.."

Answer: We apologise for the error. The text has been corrected.

Lines: 128-129: Why was PCR run on only stool samples if this was not the most sensitive method for detection of L. intracellularis (as stated in Lines: 327-328 too). Could diagnostic PCR be run on EDTA- blood samples instead for better confirmatory testing for both PCV2 and bacterial pathogens?

Answer: PCR on stool samples allowed us to select farms with presence of Lawsonia intracellularis infection. After this preliminar step we conducted the gold standard diagnostic procedure (histology and immunohistochemistry) to demonstrate the co-localization of Lawsonia intracellularis with the lesion the bacterium produces. Because Lawsonia intracellularis does not produces bacteraemia, it is not possible its detection by PCR on blood samples. This latter instead is possible in case of PCV2 infection because this virus uses bloodstream.

Section 2.2 and Section 2.5: What was the reason to run PCR on other bacteria as well as L. Intracellularis and isolate them? Why was L. Intracellularis not isolated too? Was there any correlation between the other bacteria and PCV2?

Answer: All specimens screened were from animals with a clinical diagnosis of PHE. Since the symptoms were very similar to those of acute swine dysentery and brachyspiral colitis, screening for Brachyspira hyodysenteriae, Brachyspira pilosicoli, was also performed. With respect to L. intracellularis isolation, the difficulty in routinely culturing has led to several alternative methods for confirmation of L. intracellularis infection (e.g. immunohistochemistry, in situ hybridization, PCR).

Lines 216-218: What is the significance of the age of the pigs being sampled? Are pigs under 100 days old still being weaned in which case potentially affected by maternal antibodies?

Answer: The 28 pigs examined ranged in age from 49 to 220 days old and were representative of all age groups present on fattening farms. One hundred days old is the median age of the pigs examined. Regarding maternal immunity, progeny from acutely affected breeding stock is not fully protected from PE (Jacobson et al. 2010; https://doi.org/10.1016/j.vetmic.2009.09.034).

Lines 349-351: This point summarising the involvement of PCV2 with the immune response is key in the understanding of the effect of co-infection and supports the hypothesis of the paper. The article would benefit with having this reasoning introduced and discussed at an earlier stage. The abstract did indeed mention a possible link to a co-infection but alluded to L. intracellularis being instrumental in PCV2 infection, whereas the statement discussed here implies the opposite. This is furthermore supported in Line 370 and Lines 376-381. The rationale for the methodology and the significance of the results was not immediately apparent to me until this was discussed.

Answer: even if the negative immunomodulation is the main mechanism triggered by PCV2 to allow secondary infections it is active only in PCV2 systemic infections. To date we do not know (because lesions in lymphoid tissues are lacking) if the mechanism is really operative in other PCV2 diseases. Data from literature in dual infections PCV2- Lawsonia intracellularis emphasize the role that pathogens activating macrophages have to improve PCV2 replication. For this reason this was also our conclusion in the summary. Hovewer we thank the reviewer for this suggestion. We have added text in the discussion to clarify this (L. 354-356).

Line 398: Typographical error — L. intracellularis referred to as LI.

Answer: We apologise for the error. The text has been corrected.

Reviewer 2 Report

Several papers cited by the authors in this manuscript already use real-time PCR and immuno-histochemistry (IHC) as companion methods for detecting both Lawsonia intracellularis as well as porcine circovirus type 2 (PCV2). While it is notable that PCR is the more robust screening method for L. intracellularis by a small margin, I do question what insight this adds to the current prevailing methods of detection for this pathogen.

I have several concerns with the analysis of PCV2 in this study as well. It is noted in this manuscript that PCV2 infection can cause lesions to the ileum that are macroscopically indistinguishable from L. intracellularis. An argument put forward in this manuscript is that real-time PCR detection is the more effective screening method for the presence of a pathogen, but IHC should then be employed on the positive samples as a means of verifying the causative agent. Thus, the omission of PCR screening for PCV2 is somewhat strange, given that this would be a direct way of demonstrating this argument. Furthermore, while it is already argued in the literature that the dynamics of PCV2 infections can be influenced by L. intracellularis co-infections, I do not think that IHC observations from the four instances of PCV2 co-infections constitute a large enough sample size to draw any significant conclusions about this topic that are not demonstrated in greater detail elsewhere.

MINOR COMMENTS:

-The text for scoring PCV2 IHC data (lines 196-200) differs from what is written in the S1 figure legend. Please address.

-There are numerous instances of the text referencing that the presence of PCV2 is demonstrated in a figure (lines 235-237 for Fig 5A; lines 263-264 for Fig 6), but there is no accompanying text in the respective figure legends that reinforces these observations. Please add these descriptions to the figure legends or arrows to indicate areas of interest when two different pathogens are being discussed.

-The text references a Figure 6 with subsections A, B, C and D, while the figure in this manuscript only has subsections A and B. Please address.

Author Response

Several papers cited by the authors in this manuscript already use real-time PCR and immuno-histochemistry (IHC) as companion methods for detecting both Lawsonia intracellularis as well as porcine circovirus type 2 (PCV2). While it is notable that PCR is the more robust screening method for L. intracellularis by a small margin, I do question what insight this adds to the current prevailing methods of detection for this pathogen. We agree with the rewiever but most of the available investigations in literature are on light pigs and we aimed to do similar investigation also on heavy pigs usually breed in Italy.

I have several concerns with the analysis of PCV2 in this study as well.

It is noted in this manuscript that PCV2 infection can cause lesions to the ileum that are macroscopically indistinguishable from L. intracellularis. An argument put forward in this manuscript is that real-time PCR detection is the more effective screening method for the presence of a pathogen, but IHC should then be employed on the positive samples as a means of verifying the causative agent. Thus, the omission of PCR screening for PCV2 is somewhat strange, given that this would be a direct way of demonstrating this argument.

Answer: Prevalence of Lawsonia intracellularis infection is high but this does not mean that all pigs have proliferative enteropathy. PCR can demonstate the presence of Lawsonia intracellularis in a sample, i.e. provide objective indication of infection, but it cannot demonstrate the presence of the lesion. Only Ct < 20 by qPCR can be indicative of lesion and of the immunohistochemical detection of Lawsonia intracellularis. We agree with the reviewer that due to the higher sensibility of PCR, this test should not lack in a Lawsonia intracellularis diagnostic protocol (and we included), but the definitive diagnosis aimed to demonstrate Lawsonia intracellularis as the causative agent to date includes only histology and immunohistochemistry.

Furthermore, while it is already argued in the literature that the dynamics of PCV2 infections can be influenced by L. intracellularis co-infections, I do not think that IHC observations from the four instances of PCV2 co-infections constitute a large enough sample size to draw any significant conclusions about this topic that are not demonstrated in greater detail elsewhere.

Answer: The discussion of the data produced in this study on the co-infection PCV2-Lawsonia intracellularis are inspired to what suggested on this topic by Jensen et al. (https://doi.org/10.1016/j.jcpa.2006.08.006) that had a higher caseload. We agree with the reviewer that our 4 cases are too low, but when the results agree with those of other authors, maybe the conclusion is plausible.

MINOR COMMENTS:

-The text for scoring PCV2 IHC data (lines 196-200) differs from what is written in the S1 figure legend. Please address.

Answer: We apologise for the error. The text has been corrected.

-There are numerous instances of the text referencing that the presence of PCV2 is demonstrated in a figure (lines 235-237 for Fig 5A; lines 263-264 for Fig 6), but there is no accompanying text in the respective figure legends that reinforces these observations. Please add these descriptions to the figure legends or arrows to indicate areas of interest when two different pathogens are being discussed.

Answer: we apologize but in figure 5 caption it is erroneously reported L. intracellularis instead of PCV2 and in line 264 the figure is 5C.

-The text references a Figure 6 with subsections A, B, C and D, while the figure in this manuscript only has subsections A and B. Please address.

Answer: we apologize, the figure with 4 subsections is the n. 5. The text has been corrected.

Round 2

Reviewer 2 Report

These revisions have satisfied the bulk of my earlier comments. The data are now more clearly presented in the figure legends and there is better agreement between the manuscript text and the figures. While I still maintain that the PCV2 data in this manuscript is prohibitively small for drawing any broad conclusions on its own, I appreciate that more nuance has been added in the discussion section to tie these observations to existing reports in the literature.

My remaining comments are for minor typographical errors that I found in some of the recently added passages:

-Line 260: Please capitalize "figure 6B"

-Lines 253-255 "When more than a pathogen is a cause for disease[...]". Added section needs to be broadly re-edited for grammar and punctuation.

Author Response

These revisions have satisfied the bulk of my earlier comments. The data are now more clearly presented in the figure legends and there is better agreement between the manuscript text and the figures. While I still maintain that the PCV2 data in this manuscript is prohibitively small for drawing any broad conclusions on its own, I appreciate that more nuance has been added in the discussion section to tie these observations to existing reports in the literature.

Answer: The Authors thank the Reviewer for the positive comments on the paper and for the suggestions provided.

My remaining comments are for minor typographical errors that I found in some of the recently added passages:

-Line 260: Please capitalize "figure 6B"

Answer: the text has been modified

-Lines 253-255 "When more than a pathogen is a cause for disease[...]". Added section needs to be broadly re-edited for grammar and punctuation.

Answer: We thank the reviewer for the comment. The sentence has been modified accordingly.